

# Merging Fluxgate and Induction Coil Data to produce low Noise geomagnetic Observatory Data meeting the INTERMAGNET Definitive One-second Data Standard.

Heinz-Peter Brunke[GFZ], Rudolf Widmer-Schnidrig[BFO], and Monika Korte[GFZ]

[GFZ]GeoForschungsZentrum Potsdam, Telegrafenberg, 14473 Potsdam, Germany
[BFO]Black Forest Observatory, Heubach 207, 77709 Wolfach

*Correspondence to:* Brunke (brunke@gfz-potsdam.de)

**Abstract.** For frequencies above 30 mHz the instrument intrinsic noise level of typical fluxgate magnetometers used at geomagnetic observatories can mask ambient magnetic field variations on magnetically quiet days. This is especially true for stations located at mid- and low-latitudes where variations are generally smaller than at high latitudes. INTERMAGNET has set a minimum quality standard for definitive one second data. Natural field variations referred to as pulsations (Pc-1, Pc-2, Pi-1) fall in this band. Usually their intensity is so small, that they rarely surpass the instrumental noise of fluxgate magne-
tometers. Moreover high quality magnetic field observations in the band 30 mHz - 0.5 Hz contain interesting information, e.g. for the study of ionospheric electron interactions with electromagnetic ion cyclotron (EMIC) plasma waves.

We propose a method to improve 1Hz observatory data by merging data from proven and tested fluxgate magnetometers with induction coil magnetometers into a single data stream. We show how measurements of both instruments can be combined without information loss or phase distortion.

The result is a time series of the magnetic field vector components, combining the benefits of both instruments: long term stability (fluxgate) and low noise at high frequencies (induction coil). This new data stream fits perfectly into the data management procedures of INTERMAGNET and meets the requirements defined in the Definitive One-second Data Standard. We describe the applied algorithm and validate the result by comparing power spectra of the fluxgate magnetometer output with the merged signal. Daily spectrograms from the Niemegk observatory (NGK) show that the resulting data series reveal information at frequencies above 0.05 Hz that can not be seen in raw fluxgate data.

## 1 Introduction

Conventional fluxgate magnetometers used at geomagnetic observatories are optimized towards their main purpose long-term stability. Even though they have excellent properties to measure the low frequency part of the magnetic spectrum, their noise usually surpasses the natural background field variations at frequencies higher than 30 mHz at quiet days and middle or low latitude observatories. A commonly used instrument is the FGE fluxgate magnetometer (Pedersen and Merenyi, 2016). However, the demand for low noise high frequency observatory data is increasing. Variations in the band 30 mHz - 0.5 Hz correspond to plasma waves of importance for radiation belt physics. Electrons interacting with such waves can be scattered into



the loss cone and reduce the energy content of the radiation belt. Pc 1 pulsations (f>0.1Hz) can be caused by electromagnetic ion cyclotron (EMIC) waves, being subject to many recent investigations (e.g. Shprits et al. (2016) or Usanova et al. (2014)).

A quality standard for observatory data has been defined in technical note TN6 of 2.Okt.2014 "INTERMAGNET Definitive One-second Data Standard" (Turbitt et al., 2014). To meet the INTERMAGNET standard a noise level of $10pT/\sqrt{Hz}$ at 0.1Hz

is required. This specification can hardly be met by fluxgate magnetometers currently used in magnetic observatories. Even though fluxgate magnetometers with low noise level at high frequencies have been presented (e.g. Korepanov (2007)) it is not desirable to replace approved observatory magnetometers with proven long term stability, as long term absolute stability is the main requirement for geomagnetic observatories. Moreover it is not desirable to discontinue a long standing data series.

In the frequency band 30 mHz - 0.5 Hz natural magnetic signals are relatively rare and usually of small intensity. Hence,

it is often termed "dead band" in magnetotellurics. Such signals disappear in the noise because of their small intensity. In order to overcome this "blind spot" of magnetometers currently in use, we propose to improve the quality of observatory data by merging fluxgate data with induction coils data. Induction coils have very little noise, but do not at all provide long term stability. Here we present a method how to numerically merge data of both instruments. The result is a single time series combining the benefits of both instruments: long term stability (fluxgate) and low noise at high frequencies (induction coil).

The result can be understood as one second fluxgate data, noise filtered without loss of information on phase or amplitude. The resulting data product fitts perfectly in the INTERMAGNET data processing scheme established at magnetic observatories. It exceeds the INTERMAGNET noise requirement of $10pT/\sqrt{Hz}$ at 0.1 Hz by far. As a consequence three decimal digits of the merged data are valid ($1pT$). This is one digit more than required in the "INTERMAGNET Definitive One-Second Data Standard". The third digit is needed to describe weak magnetic signals that can now be observed. Otherwise the effect of

quantisation noise can be seen in the spectrum.

## 2   Noise Considerations

The instrument inherent noise of a fluxgate magnetometer can be identified in the data as white noise for frequencies above 30 mHz (fig. 3). At a normal day the part of the natural signal with frequencies above 30 mHz vanishes in the noise. Korepanov et al (Korepanov, 2007) show a typical power spectral density of the natural field at the Dourbes magnetic observatory (DOU)

and compare it to noise of their new magnetometer kept in a shielding box (fig. 1). As a green line we added to this plot the spectral noise density of the FGE in the frequency band 30 mHz - 0.5 Hz, as observed at the Niemegk observatory (NGK) and shown in fig. 3. We marked the INTERMAGNET requirement of $10pT/\sqrt{Hz}$ at 0.1Hz as a blue dot and added the noise of an induction coil magnetometer (MFS05, METRONIX) after Pulz (2010) as a red line. Whereas the fluxgate spectral noise density remains constant at high frequencies, the spectral noise density of an induction coil decreases towards higher frequencies by

20dB (factor 10) per decade up to a frequency of 100Hz (Pulz, 2010). At 1Hz its spectral noise density is at least 2 orders of magnitudes lower than the noise of any fluxgate.



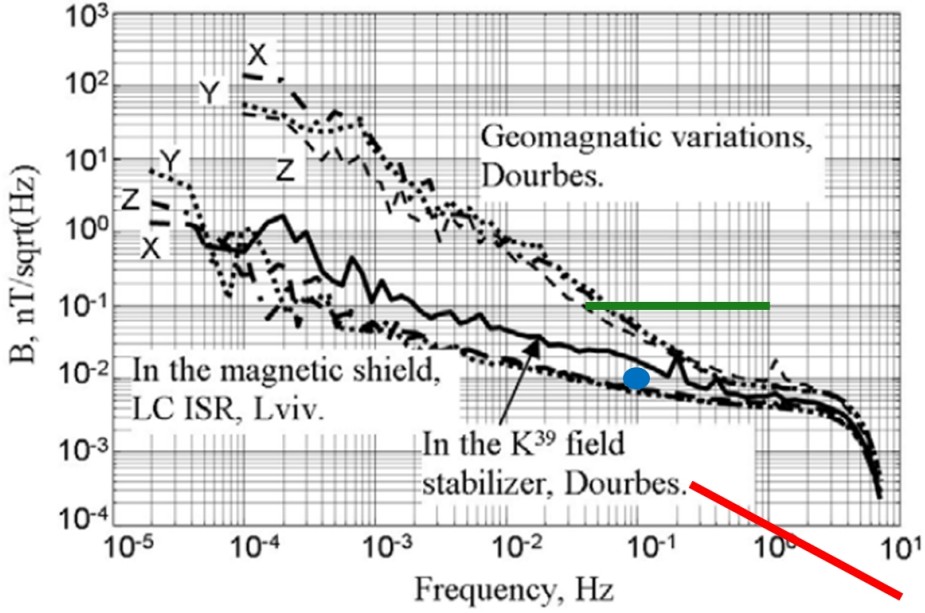

**Figure 1.** Noise spectra of a magnetometer prototype and of measured natural geomagnetic variations (modified after Korepanov (2007)). We added the high frequency noise level of the FGE magnetometer as measured at NGK (green line) and the high frequency noise of a typical induction coil magnetometer (red line). The blue dot shows the specification given in the "INTERMAGNET Definitive One-second Data Standard" (Turbitt et al., 2014).

Instrument noise at very long periods from days to years are equivalent to instrument long term stability. This is the most important quality feature of an observatory fluxgate magnetometer. As described in the following, merging data of both instruments does not in any way effect the long term stability.

## 3 Combining fluxgate and induction coil magnetometer

5    We take advantage of the fact that the signal of an induction coil magnetometer is proportional to the time derivative of the magnetic field component along the axis of the coil. In the following we consider exemplary the $X$-component. For each sample time $t_0$ we look at values $X_{FG}(t_i)$ registered by the fluxgate magnetometer at the times $t_i$ varying from $t_0 - T_s N$ to $t_0 + T_s N$, $T_s$ being the sample width. Additionally we need recordings of the induction coil also at the times $t_i$. We get as result of our algorithm the value $B_X(t_0)$ as a more precise value of the magnetic field at the time $t_0$.

10    Integrating induction coil data provides a curve which is comparable to fluxgate data. The integration constant $B_X(t_0)$ is the field at the start time $t_0$ of the integration. Due to the low noise content of an induction coil this curve is smooth compared to fluxgate data (blue line in fig. 2). Our approach is, to fit the former like a spline curve to the fluxgate data (fig. 2 , right). This effectively eliminates instrument noise of the fluxgate by averaging. However, the shape of the fitted curve results from





the induction coil measurements and the loss of information due to averaging is replaced by information stemming from the induction coil.

For given time $t_0$ and a variable time $t_i$ in its environment, data of fluxgate $X_{FG}$ and induction coil magnetometer $\dot{X}_{IC}$ can be expected to satisfy the following equation, assuming that both sensors are aligned with the X-axis:

$$X_{FG}(t_i) - B_X(t_0) = \int_{t_0}^{t_i} \dot{X}_{IC}(t)dt \qquad (1)$$

Assuming that the induced voltage measured over the induction coil is object to scale factor $C$ and offset $\Delta U$:

$$X_{FG}(t_i) - B_X(t_0) = \int_{t_0}^{t_i} (C \cdot U_{IC}(t) + \Delta U)dt \qquad (2)$$

The blue line in fig. (2, left) is achieved evaluating the integral from the fixed central time $t_0$ to a time $t_i$ with $i$ varying from $-N$ to $+N$ applied to an actually measured induction coil data set. The according time series of a flugate magnetometer is given as green line.

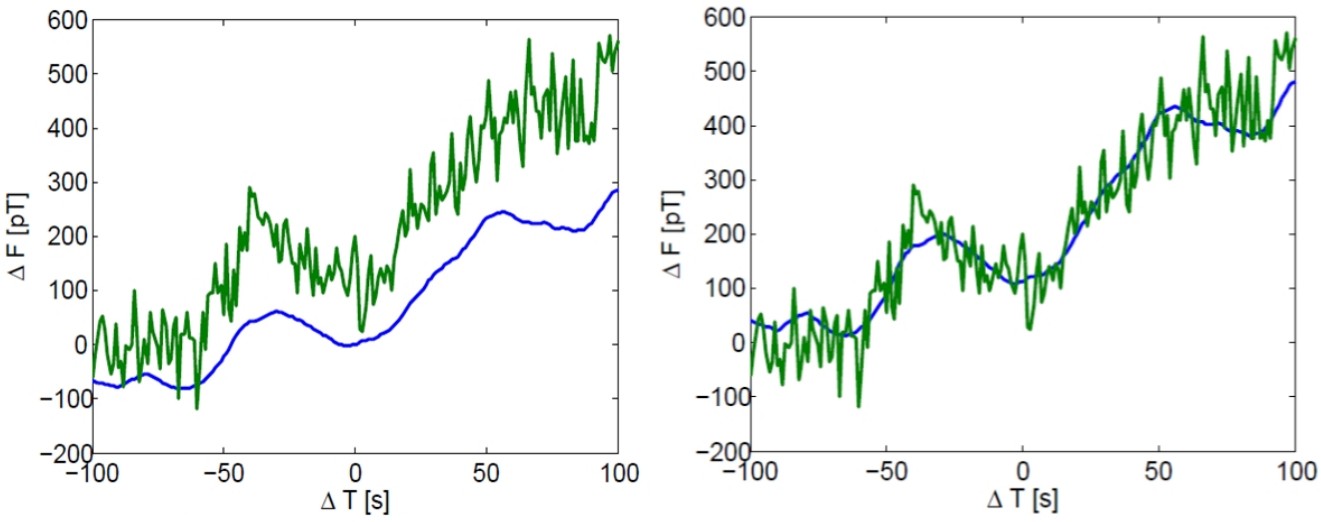

**Figure 2. (left):** Fluxgate data (green) and integral of induction coil signal (blue). **Right:** Blue curve fitted to FG data by adapting $C$, $\Delta U$ and $B_X(t_0)$ in equation 2.

The blue line (IC) is much smoother than the green one (fluxgate data). Choosing the right values for the parameters $\Delta U$, $C$ and $B_X(t_0)$ it can be fitted like a spline to the fluxgate data, as shown in the right panel of fig. 2. For this fit, equation (2) has to be solved for $\Delta U$, $C$ and $B_X(t_0)$ as unknowns. Equation (2) delivers with fixed $t_0$ and varying $t_1$ one equation for each $t_1$ to determine these unknowns. As shown in the figure (2) we use a time interval of 200sec. The time spread of 200sec was determined empirically. A longer time interval increases the number of equations and therefore might improve the





numerical accuracy of the result. But the time interval must be small enough to ensure not to distort the fluxgate signal at lower frequencies. In chapter 4 we show, that a further improvement is not necessary because the signal is clearly resolved up to the Nyquist frequency of 0.5Hz.

Assuming that both data series are available at a sample rate of 1Hz and approximating the integral by a sum, equation 2 can be written in the discrete form

$$\sum_{t_0}^{t_i} (C \cdot U_{IC}(t) + \Delta U) T_s - B_X(t_0) = X_{FG}(t_i) . \tag{3}$$

Assuming that $T_s = 1\text{sec}$, one equation can be derived for each sample time $t_1$. If $i$ varies from $-N$ to $+N$ we get the following conditional system for the parameters $C, \Delta U$ and $X_{FG}(t_0)$:

$$
\begin{aligned}
C \cdot S_N &\quad + \Delta U \cdot N &\quad + B_X(t_0) &\quad = X(t_N) &\quad + r_N \\
&\qquad\qquad ... \\
C \cdot S_i &\quad + \Delta U \cdot i &\quad + B_X(t_0) &\quad = X(t_i) &\quad + r_i \\
&\qquad\qquad ... \\
C \cdot S_{-N} &\quad + \Delta U \cdot (-N) &\quad + B_X(t_0) &\quad = X(t_{-N}) &\quad + r_{-N}
\end{aligned}
$$

This is a system of linear equations of the Form $G \cdot x = y$. With $S_i = T_s \sum_{k=0}^{i-1} U_{IC}(k)$ and written as explicit matrices:

$$
\begin{pmatrix}
S_N & N & 1 \\
& \vdots & \\
S_i & i & 1 \\
& \vdots & \\
S_{-N} & -N & 1
\end{pmatrix}
\cdot
\begin{pmatrix}
C \\
\Delta U \\
B_X(t_0)
\end{pmatrix}
=
\begin{pmatrix}
X(t_N) \\
\vdots \\
X(t_i) \\
\vdots \\
X(t_{-N})
\end{pmatrix}
\tag{4}
$$

Setting $N$ to 100 we get a time interval of 200sec leading to 201 equations. Obviously we have more than three conditional equations at hand and a solution must be found in the sense of minimizing the mean square of the residuals. The so called normal equations (Schmucker and Weidelt, 1975) are $x = (G^T \cdot G)^{-1} \cdot (G^T \cdot y)$. More explicit for this case:

$$
\begin{pmatrix}
C \\
\Delta U \\
B_X(t_0)
\end{pmatrix}
=
\begin{pmatrix}
\sum_{i=-N}^{N} S_i^2 & \sum_{i=-N}^{N} i \cdot S_i & \sum_{i=-N}^{N} S_i \\
\sum_{i=-N}^{N} i \cdot S_i & N(N+1)(2N+1)/3 & 0 \\
\sum_{i=-N}^{N} S_i & 0 & 2N+1
\end{pmatrix}^{-1}
\cdot
\begin{pmatrix}
\sum_{i=-N}^{N} S_i X(t_i) \\
\sum_{i=-N}^{N} i X(t_i) \\
\sum_{i=-N}^{N} X(t_i)
\end{pmatrix}
\tag{5}
$$

$B_X(t_0)$ is the field value at the time $t_0$ we are looking for. Note that the numeric effort for this calculation is very small. It is restricted to the inversion of a 3x3 matrix and calculating six sums in equation (5) for each sample of the resulting time series. The sums do not even have to be evaluated for each new $t_0$. They can be updated from the previous time step.





## 4   Results

The success of merging both instruments can best be demonstrated by comparing power spectra of unprocessed FGE and merged data and looking at spectrograms of the respective data series. In figure (3) power spectral densities (PSD) of fluxgate data (FGE) as measured in NGK are compared to PSD of merged data. The instrument inherent noise can clearly be identified

5   in the unprocessed data as the white noise (constant noise level) of $0.01 nT^2/Hz$ for frequencies above 40 mHz. At the same frequencies the PSD of the merged data show a continuing linear decay as expected for the natural field. Very little influence of noise is visible in the PSD of the merged data. At frequencies lower than 10 mHz the spectra of FGE and merged data are identical.

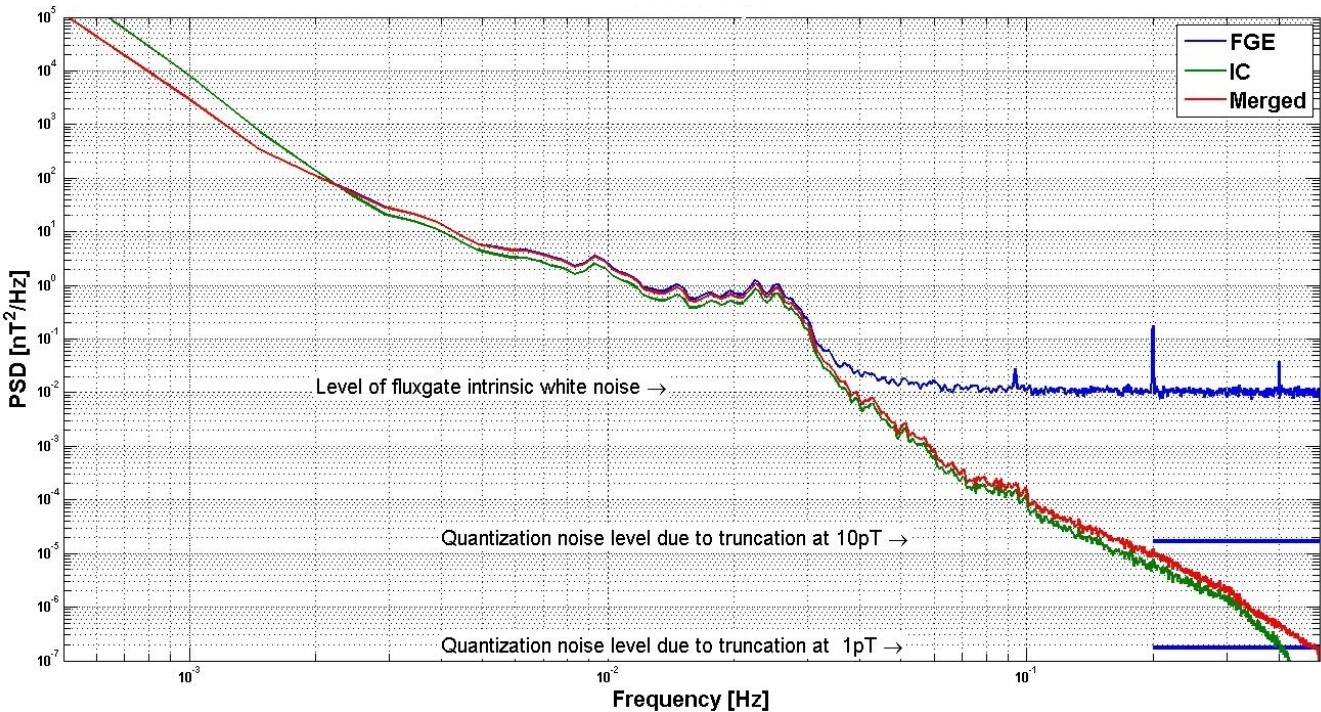

**Figure 3.** Spectra of unprocessed (FGE) and merged data of the entire day 17.July2016. The level of the fluxgate inherent white noise is clearly visible for frequencies over 0.04Hz (blue), whereas the merged data show the continuing decay as expected for the natural field (red).

As shown in figure (4) the coherence between merged data and FGE data is perfect at low frequencies (< 0.01Hz). At high

10   freqencies (>0.03Hz) the coherence is very good between merged data and IC data. It is not perfect due to the low energy content of the signal at these frequencies. A very high coherence is also observed between FGE and IC data at frequencies between 0.003Hz and 0.03Hz. In this frequency band both instruments are best comparable, allowing to check alignment and intercalibration of both instruments. Induction coils are usually run at a GPS synchronized data logger. This allows to verify the correct time stamping of the FGE data logger as proposed in the "INTERMAGNET Definitive One-second Data Standard"




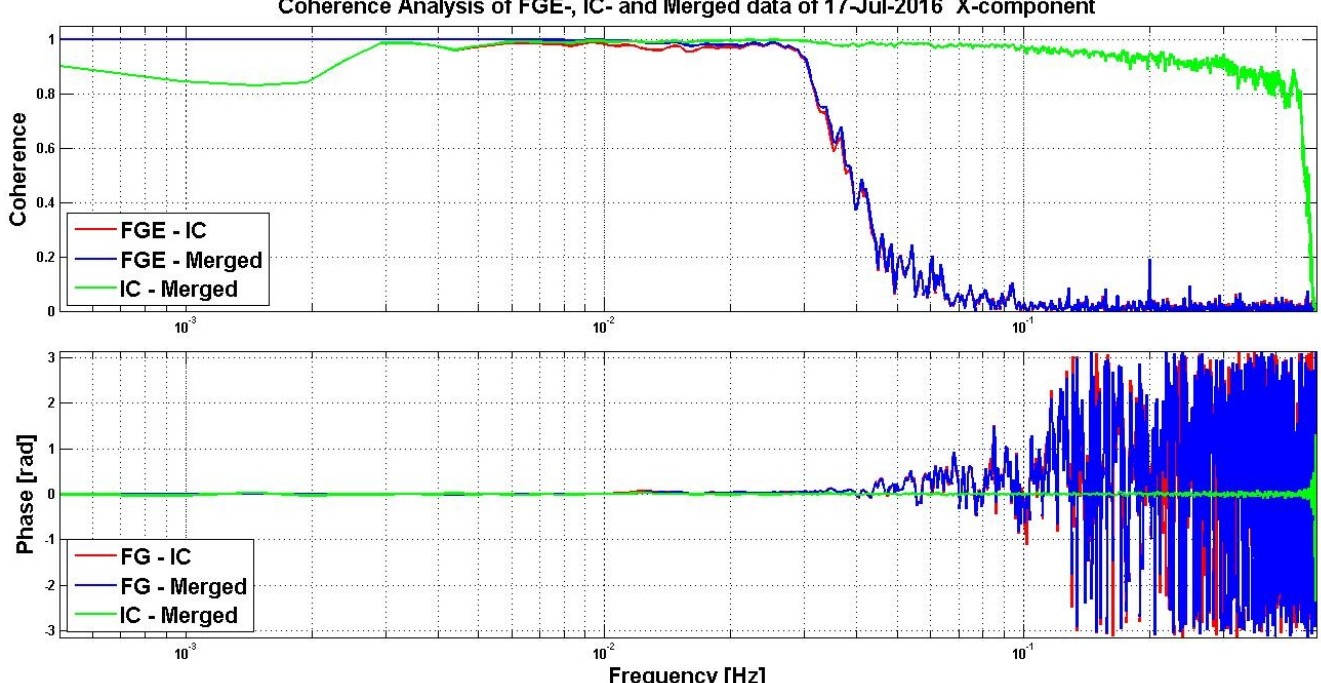

**Figure 4.** Coherences between FGE, merged data and induction coil data (IC). A perfect coherence between merged data and FGE data (blue) is observed at low frequencies (< 0.01Hz). At high freqencies (>0.03Hz) the coherence is high between merged data and IC data (green).

(Turbitt et al., 2014). For such a test a natural or artificial signal clearly surpassing the FGE white noise level is required. Though a time stamp error of 0.01sec can hardly be verified on a data series sampled at 1/sec which is accordingly Nyquist filtered.

Figure (5) shows spectrograms for three selected days. The spectrograms are overlain with time series of the field (black line). Panel (a) and (c) are directly comparable as both cover the 2 Feb. 2016. Panel (a) is based on merged data, (c) on fluxgate data only. Whereas in (a) the spectral content is clearly resolved up to the Nyquist frequency of 0.5Hz in panel (c) only the strongest activity surpass the noise level in the frequency band from 30 mHz up to 0.5 Hz. These are possibly field line resonances and their lowering frequencies towards the night time is clearly visible in the merged data in panel (a). Panel (a) also shows with help of the overlain magnetogram that irregular pulsations (Pi pulsation, no clear frequency) are related to the beginning of substorms. Panel (b) shows around 18:00 an example of a pulsation event with a clearly defined center frequency of about 300 mHz (Pc-1 pulsation) and panel (d) shows an example of a pulsation whose main frequency rises from about 20 mHz at 15:30 to 400 mHz at 16:30. Additionally, panel (b) shows horizontal structures from 6:00 to 17:00 (16.2sec square wave) and sweeps with falling frequency starting at 6:30 and 9:00. Both are man-made contamination. We found a lot of other remarkable signals revealed in such spectrograms, but discussing them is beyond the scope of this paper.





**Figure 5.** Spectrograms of three selected days. Panels (a) and (c) allow for a direct comparison between merged data (a) and fluxgate only (c). Interesting are the irregular pulsations related to substorm activity in (a), pc-1 pulsations with clearly defined frequency of 300 mHz in panel (b) at 18:00 and in (d) pulsation activity with rising frequency starting at 15:30 potentially related to EMIC waves.



## 5 Discussion

Providing data to a user is generally linked to the commitment to the best possible accuracy. Speaking in terms of data series, this means to optimize the signal to noise ratio over the entire available band width. Hence our effort improves magnetic data in view of the INTERMAGNET one sec standard. We are well aware, that merging two data sets does not produce new

information. Merging data from different instruments at geomagnetic observatories is not a far-fetched idea. On the contrary, it is already standard technique when definitive data are produced by combining absolute observations from DI-theodolite and scalar magnetometer with vector variometer (fluxgate) data. As INTERMAGNET does not cover induction coil data, applying our method makes additional information available to the INTERMAGNET community. Plots as presented in fig. 5 could also be produced using both data sets separately. But after applying our method the single one-second data series as administered

for many observatories worldwide by INTERMAGNET is sufficient. The effort of storing and exchange of data sampled at 1/sec as proposed by INTERMAGNET makes more sense if the entire possible bandwidth contains information about the Earth magnetic field. However, in order to take full advantage of the high frequency information, the data should be reported to at least three decimal digits (1 pT). Otherwise quantization noise becomes visible. The quantization noise level due to truncation is shown in figure (3).

The INTERMAGNET one second standard prescribes a hardware low-pass filter with a corner frequency of 0.2 Hz for the fluxgate data. This filter has absolutely no effect to the result of the merging process. The high frequency information stems from the induction coil side only. We verified this by numerical experiments.

The induction coil data should better be sampled at a rate higher than 1/sec and subsequently been filtered and down sampled numerically in order to best avoid aliasing into the Nyquist bandwidth of the fluxgate data. Variometer data are often explored in

the time domain. The morphologies of events like sudden storm commencements, sub-storms or pulsations are investigated by visual or automated inspection. In merged data events with even very small amplitudes can be observed, as noise is eliminated to a large extent.

It suggest itself to benefit from the possible high sample rate of an induction coil and to extend the bandwidth up to the 50Hz grid frequency limit. The method can easily be extended accordingly. But this is beyond the scope of this paper.

We have to assume some preconditions for the application of our method:

- We made the assumption, that the sensor of the fluxgate and the induction coil magnetometer are perfectly aligned. If this is not the case, the induction coil signal will be affected by the other components. The fitting of both data will become worse. This could be accounted for by numerically rotating the data beforehand. For the determining the rotation matrix a larger data set is needed. But induction coils can be used down to relatively low frequencies, so that the overlap with

the fluxgate is broad enough to determine the misalignment.

- We assume that the induction coil output is proportional to the time derivative of the magnetic field. Induction coils normally have a ferrite core and use electronic amplification (Pulz, 2010). Both can influence the linear transfer function in the used frequency band from 30 mHz to 1 Hz. This effect could be compensated by an according numeric amplitude pre-filtering. This point is especially important, if the method shall be extended to higher frequencies.



In the process of curve fitting depicted in fig. 2 it is a good idea to weight points in the direct environment of the time of interest $t_0$ higher than remoter ones. We made good experience with a Gaussian filter. For the sake of simplicity we did not include weights in the equations (3 to 5).

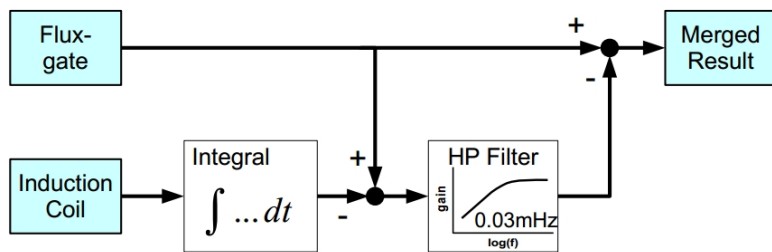

**Figure 6.** Data flow of a filtering method to merge fluxgate data and induction coil data. The cut-off frequency of the low-pass filter should be 30 mHz. Calibration of the induction coil data is not included.

The method we described above is inspired by the quite descriptive process of curve fitting in the time domain. Another way to do the merging is a filtering method (See fig. 6 for a depiction of the data flow.). The difference between fluxgate and integrated induction coil data is calculated and subsequently high-pass-filtered. The obtained intermediate result contains the high frequency content of the fluxgate data (mainly noise) and the inverse of the high frequency signal content of the induction coil. In the intermediate result drift of induction coil data and the integration constant is eliminated by the high-pass-filter. In a final step the intermediate result is subtracted from the fluxgate data to get the merged output. The output contains the low-pass-filtered fluxgate data (the high-pass content has been subtracted) and the integrated, high-pass-filtered induction coil data. That means, the high frequency content of the fluxgate, containing mainly noise, is eliminated in the resulting data and replaced by the according high frequency part of the induction coil side.

An advantage of the curve fitting method is, that the scale factor of the induction coil is automatically corrected. The numerical effort of the curve fitting method may be slightly higher. But the effort is restricted to solving one 3x3 matrix for each data sample. Extending the curve fitting method to a higher sampling rate of the induction coils (e.g. up to 50Hz) is straightforward.

## 6 Conclusions

We have shown how data of fluxgate magnetometer and induction coil data can be merged. A numerical method based on the idea of fitting integrated induction coil data to fluxgate data has been presented in detail. The merged data set combines the excellent long-term stability of well-proven observatory fluxgate magnetometers with the low noise content of induction coil magnetometers at high frequencies. The resulting merged data differ from unprocessed fluxgate data only within the range of the intrinsic fluxgate noise (<0.1nT). Hence the long term stability is in no way affected. We produce a data set that has the format of usual one second fluxgate data but surpasses the specification of the "INTERMAGNET Definitive One Second Data



Standard" by far. The resulting data set reveals details like Pc-1, Pc-2 and Pi-1 pulsation of even small intensities which are not resolved by a fluxgate magnetometer. The spectral noise density in the resulting data set is up to the Nyquist of 0.5Hz clearly lower than the spectral density of the natural magnetic field even at days with low activity. Thus with our method we can fill the entire possible bandwidth resulting from the "INTERMAGNET Definitive One Second Data Standard" with meaningful information about the Earth magnetic field.



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
