# Peer review of "Merging Fluxgate and Induction Coil Data to produce low Noise geomagnetic Observatory Data meeting the INTERMAGNET Definitive One-second Data Standard."

_Geoscientific Instrumentation, Methods and Data Systems, 2017_

## Referee Comment (RC1) · S. Khomutov (Referee) · 10 Jun 2017

Comments to **Merging Fluxgate and Induction Coil Data to produce low Noise geomagnetic Observatory Data meeting the INTERMAGNET Definitive One-second Data Standard** of Heinz-Peter Brunke, Rudolf Widmer-Schnidrig, and Monika Korte

The article presents an approach for improving 1-sec Definitive data, which allows to overcome the
5   problem of the intrinsic noise of fluxgate magnetometer at frequencies above 0.03 Hz. The authors propose to use the data of induction magnetometer, which do not have these limitations (intrinsic noise) at high frequencies, combining them with the data of fluxgate magnetometer. The proposal is very interesting and, as the authors show, can be effective and promising.

I think that the article can be published, however, there are some issues that require more careful
10   consideration:

1) the description of the technology of computing the merged data in Section 3 gives an understanding of the individual steps of the procedure, but, in my opinion, does not allow us to understand the chain of transformations as a whole.

I understand the processing of data as follows: we have daily set of 1-sec values $Xf(t)$ of variations
15   of component X, recorded by fluxgate magnetometer. Also we have similar set of derivation $dXi(t)/dt$ of X-component, recorded by induction magnetometer. We need to calculate the merged values $Xfi(t)$ over all day.

Consider, as example, calculation for time $t0=01:30:00UT$ – we need to get value of $Xfi$ for this time. Authors use the set of both magnetometers with length of 200 sec, i.e. from 01:28:20 to
20   01:31:40 and estimate the parameters C, deltaU and Bx, which are constant over this interval. But next step, i.e. the calculation of $Xfi(t0)$, is not clear. It can be suggested that this value is calculated using equation (3) or this value is value $Bx(t)$ in according to P3,L09. Then we go to text time $t0=01:30:01UT$ and the calculations are repeated.

I think that authors need to explain this question more clearly.

25   2) the authors write about low noise of induction magnetometers at frequencies above 0.03 Hz (P2, L12) and Figure 1. However, this is only about our intrinsic noise. At the same time, a number of factors are known that increase the noise of induction magnetometer in real measurements. For example, this is the sensitivity to mechanical oscillations of the sensors (microseisms, earthquakes, vibrations, etc.) or interference in long lines with an analog signal. The latter can be especially
30   unpleasant, since it gives a noise to differential (measured) signal. In the integration, such noise, even localized in time (spike), can give an effect throughout the entire integration interval, and thus also affect the low-frequency range of the merged signal.

I think that the authors should note these questions in the article, since many observatories are located in areas with increased industrial noise, effected the induction magnetometer records.

35   Also there are some specific questions:

P2,L16 – fits?

P2,L17 – way of writing of "10pT/pHz" is differed from similar text at P2,L04 (normal and Italic

fonts)

P2,L23 – I don't sure, that rule of GI allows to make the reference to Figure 3 before first references to Figure 1 and Figure 2

P3,L08 – may be Ts is half of sample width?

5    P3,L11 - "the start time t0 of the integration" – if t0 is start of integration, then why do we need data before t0? May be two integrals are used, with limits of integration (ts, t0) and (t0, ts) for -N and +N, accordingly?

P3,L13 – perhaps the reference "(blue line in fig. 2)" is best given immediately after "this curve" (P3, L12)?

10    P3,L12 – It is unclear why splines are mentioned

P3,L14 – what is "This"?

P4, Figure 2 – axis of ordinate is signed as deltaF, but in text there is no any mentions about deltaF

P4,L13 – it makes no sense to refer to splines?

P4,L13 – does t(i) need instead of t(1)? Also at P4,L14 and P5,L07

15    P5,L07 - "parameters C, deltaU" – is the Bx(t0) also parameter?

P5,L08 – there is no explanation of $r_N$

P5,L14 – authors write, that solution is least-square method. Why mention splines at P3,L12 and P4,L12?

P6,L02 – it would be nice to also show a time sets of these records.

20    P6,L04 – what is model of induction magnetometer is used for data?

P6,L04 – authors often write about the model FGE of the fluxgate magnetometer as a general type, but for induction magnetometer this is not. May be it can use "MFS05" similar to "FGE"?

P8, Figure 5 – it would be very useful to see a comparison of spectrograms of fluxgate and merged data for May 3, 2016, in order to estimate the effects of pulsations near 18UT.

---

## Short Comment (SC1) · 14 Jun 2017

Dear Sergey,

Thank you for your valuable hints and corrections. My replays are as follows:

Issue 1) We did add the sentence "The method works comparable to a filter. It acts on the measurements at the times $t_i$ varying from $t_0-T_s N$ to $t_0+T_s N$, and produces one output value for the time $t_0$. For each new $t_0$ the entire process has to be repeated using a new set of measurements."

Issue 2) As shown in Fig.(3) the combined data have up to five orders of magnitude less noise than normal FGE data. It is clear, that such data are more susceptible to environmental noise. Figure (5,b) shows such artefacts. In data from another observatory we could see the Polarization currents of a proton-magnetometer. But also Fluxgate data may contain noise, which is eliminated by the combination with induction coils. Figure (5,c) shows an Example (signal content at 0.2Hz). Please note, that the long term stability is not at all influenced by such noise effects. The integration time is always restricted to the width of the environment around t_0 which is from −100sec … +100sec. Please note also that the changes that we apply to the data are within the noise level of a normal fluxgate (+-0.1nT).

I tried to follow your comments and recommendations if ever we could. But we have the following comments:

P3,L08:  Ts is the time between two Samples. The time-window of used IC data spans from −N*Ts to +N*Ts

P3,L11: The Integration starts at t0 and goes to ti. Yes, ti can be lower than t0. This is a bit particular, but mathematically absolutely correct.

P6,L02: We discussed to show a time set. But a time set spreads over 24h in time direction and, if I remember well, about 30nT X direction. The integration is maximal 100sec long and the effect of our method to the fluxgate data is within the noise of 0.1nT. In both dimensions a time series can not enlighten anything of interest with respect to our method. So we decided not to show the time series of the data.

P6,L04: Induction coils are not a known brand. They have been made about 30years ago in the Niemegk workshop. Also the data logging is a bit outdated. We sample the induction-coil data at 1sec only. It is planned to install a more up-to-date logger and to use a higher sample rate. The test data for the paper stem from the old system.

P8,Figure 5: Yes, there are many interesting spectrograms within this framework. We decided to restrict it to only four in order not to overburden the paper. I included the respective plot here. The Pulsation around 18:00 almost vanishes in the fluxgate noise.

Your effort is really appreciated. It would be good, if you could further contribute to the discussion, in case there are still open questions or ideas to improve the article.

Best regards, Heinz-Peter Brunke

[Figure]

FGE Data 03-May-2016  Y-component

---

## Referee Comment (RC2) · Anonymous Referee #2 · 26 Jul 2017

The manuscript has been revised for minor corrections. Points raised have been addressed. The paper can be accepted for publication
* * *

---

## Author Comment (AC1) · 27 Jul 2017

**Reply to anonymous referee 3:**

**Comment of referee:** "The manuscript has been revised for minor corrections. Points raised have been addressed. The paper can be accepted for publication"

**Reply:** Thank you for the effort to read the article and to check if raised points have been addressed in a satisfactory way.

At this place I would also like to express my thankfulness also to Lars Pedersen and Sergey Khomutov who also made the effort to read, and correct the manuscript. You gave a lot of valuable hints and very constructive comments. Many errors could be corrected and misleading wordings could be improved for a better understandability.

All referee-comments and our replies are traceable in the peer review part of the online publication.

Nevertheless I want to reproduce one comment of Lars here. Lars is the supplier of the FGE instrument. I consider his comment very important, because it shows that our paper must not be misunderstood in any way as criticism to the FGE instrument.

**Comment of Lars:** "It makes perfect sense to combine fluxgate and induction coil data, since it can 'improve' the fluxgate data quality above 30 mHz. And I guess that merged data also above 1 Hz (like 10 Hz data) will be useful for many purposes.
But if data shall follow the INTERMAGNET standard for 1 second data, they have to be low-pass filtered starting at 0.2 Hz, and this means that there may not be any useful signal left above 0.3 Hz. And then it may be difficult to find pulsations even in the merged data. But it will still be a big improvement."

**Our Reply:** Thank you for the flowers! Yes, literally spoken a 0.2Hz low-pass filter is required by the INTERMAGNET Standard. But that is more a feature of the INTERMAGNET standard than a deficiency of our method. As the spectrograms of actual data show, there is absolutely no need to apply such a filter to the merged data. This is another slight "reinterpretation" of the INTERMAGNET standard in addition to the tree digits of resolution we propose. We added in the discussion: "However, in order to take full advantage of the high frequency information, the data should reported to three decimal digits (1 pT) and the 0.2Hz low-pass filter prescribed by the present INTERMAGNET one second standard does not have to be applied. "